# An investig-ation into the epidemiology of chikungunya virus across neglected regions of Indonesia

**Samuel C. B. Stubbs**[1¤*], **Edison Johar**[2], **Frilasita A. Yudhaputri**[2], **Benediktus Yohan**[2], **Marsha S. Santoso**[2], **Rahma F. Hayati**[2], **Dionisius Denis**[2], **Barbara A. Blacklaws**[1], **Ann M. Powers**[3], **R. Tedjo Sasmono**[2], **Khin Saw Aye Myint**[2*], **Simon D. W. Frost**[1,4¤]

**1** University of Cambridge, Department of Veterinary Medicine, Cambridge, United Kingdom, **2** Eijkman Institute for Molecular Biology, Jakarta, Indonesia, **3** Division of Vector-Borne Diseases, Centers for Disease Control and Prevention, Fort Collins, Colorado, United States of America, **4** Microsoft Research, Redmond, Washington, United States of America

¤ Current address: London School of Hygiene and Tropical Medicine, London, United Kingdom
* sam.stubbs@lshtm.ac.uk (SCBS); khinsawying@hotmail.com (KSAM)

**Data Availability Statement:** All consensus sequences generated and analysed for this study were uploaded to NCBI Genbank under accession numbers MT591083 - MT591107. Multiple

## Abstract

### Background

Chikungunya virus (CHIKV) is an important emerging and re-emerging public health problem worldwide. In Indonesia, where the virus is endemic, epidemiological information from outside of the main islands of Java and Bali is limited.

### Methodology/Principal Findings

Four hundred and seventy nine acutely febrile patients presenting between September 2017–2019 were recruited from three city hospitals situated in Ambon, Maluku; Banjarmasin, Kalimantan; and Batam, Batam Island as part of a multi-site observational study. CHIKV RNA was detected in a single serum sample while a separate sample was IgM positive. IgG seroprevalence was also low across all three sites, ranging from 1.4–3.2%. The single RT-PCR positive sample from this study and 24 archived samples collected during other recent outbreaks throughout Indonesia were subjected to complete coding region sequencing to assess the genetic diversity of Indonesian strains. Phylogenetic analysis revealed all to be of a single clade, which was distinct from CHIKV strains recently reported from neighbouring regions including the Philippines and the Pacific Islands.

### Conclusions/Significance

Chikungunya virus strains from recent outbreaks across Indonesia all belong to a single clade. However, low-level seroprevalence and molecular detection of CHIKV across the three study sites appears to contrast with the generally high seroprevalences that have been reported for non-outbreak settings in Java and Bali, and may account for the relative lack of CHIKV epidemiological data from other regions of Indonesia.

sequence alignment files, BEAST XML, log and tree files are available at: 10.5281/zenodo.3891450. Raw sequence data have been deposited in ENA under study PRJEB39967.

**Funding:** This study was co-funded by the UK Medical Research Council (MR/P017541/1) held by SDWF and the Indonesian Science Fund (DIPI) and Indonesia Endowment Fund for Education (LPDP) held by RTS (MRCUK6) as part of the UK-Indonesia Joint Health Research Call on Infectious Diseases (2016). SDWF was supported in part by The Alan Turing Institute via an Engineering and Physical Sciences Research Council grant (EP/510129/1). US-CDC supports the Emerging Virus Research Laboratory within the Eijkman Institute for Molecular Biology. The findings and conclusions in this report are those of the authors and do not necessarily represent the views of the Centers for Disease Control and Prevention. The funders had no role in study design, data collection and analysis, decision to publish, or preparation of the manuscript.

**Competing interests:** I have read the journal's policy and the authors of this manuscript have the following competing interests: Commercial funder Microsoft Research provided financial support in the form of a gift to the London School of Hygiene and Tropical Medicine, intended to support salary for SCBS. SDWF is a paid employee of Microsoft Research, which provided support in the form of salary. Microsoft Research had no role in the study design, data collection and analysis, decision to publish, or preparation of the manuscript.

## Author summary

Outbreaks of chikungunya virus (CHIKV) are a common occurrence in Indonesia. However, limited data is available on CHIKV from regions outside of the main, central islands of Java and Bali. We recruited hospital patients from three cities located in the east (Ambon), west (Batam) and north (Banjarmasin) of the country, and screened their blood for evidence of CHIKV infection. Our results showed that CHIKV infections were relatively uncommon across patients from all three sites, suggesting that CHIKV transmission is currently relatively rare in these regions.

Additional analysis of 25 recent Indonesian CHIKV genome sequences revealed that a new lineage of CHIKV has recently emerged in Indonesia. Several reports have highlighted Indonesia as a major source of imported CHIKV cases, suggesting that this new lineage has the potential to be introduced into neighbouring countries in the near future, with unknown consequences. Overall, our results indicate that additional CHIKV surveillance studies in Indonesia and Southeast Asia are needed in order to gain a clearer understanding of transmission routes and hot spots throughout the region.

## Introduction

Chikungunya virus (CHIKV) is a zoonotic pathogen belonging to the *Alphavirus* genus of the family *Togaviridae*. The virus is arthropod-borne, transmitted to humans through the bite of infected *Aedes* species mosquitoes. Phylogenetic analysis has demonstrated that CHIKV can be divided into three genotypes historically defined by their distinct geographical regions: the West African genotype, the Asian genotype, and the East, Central and South African (ECSA) genotype [1]. In recent years, 2 of these genotypes have spread into new areas. The ECSA genotype has given rise to the Indian Ocean lineage (IOL), which emerged in 2004 and rapidly spread across the islands of the Indian Ocean, India and South-East Asia [2]. In 2013, the Asian genotype, endemic to many countries in Southeast Asia, was introduced to the Americas, giving rise to major epidemics throughout the continent [3] and has since become established in the region.

Sporadic outbreaks of Asian genotype CHIKV have been reported in South and Southeast Asian countries including Thailand, Malaysia, Indonesia and the Philippines since the 1950s [4]. Following the widespread introduction of ECSA strains into Asia, outbreaks of both genotypes have been reported throughout the region, including co-circulation of the two in Indonesia, Thailand, the Philippines and Malaysia. In Southeast Asia, CHIKV is primarily transmitted by the urban mosquito species *Aedes aegypti*, which maintains the virus in a mosquito-human-mosquito cycle [5]. CHIKV is endemic to many countries in this region, including Indonesia, Malaysia, the Philippines, Singapore, Thailand and Vietnam, where it is continuously maintained through persistent low-level transmission in the local population [4]. This is punctuated by larger, epidemic outbreaks, which occur at irregular intervals of between 2–20 years [6]. It is also postulated that CHIKV may be maintained in the mosquito population by vertical and horizontal transmission, both processes that have been demonstrated experimentally [7,8] but not confirmed in the field. The existence of a mammalian reservoir host has also been suggested, and antibodies against the virus have been detected in macaque populations in both the Philippines [9] and Malaysia [10], suggesting wildlife may be involved in maintaining the virus between epidemics. However, their role in maintenance of the virus is

as yet unproven as an earlier survey of macaques in Sri Lanka revealed all to be serologically negative [11].

In Indonesia, outbreaks of CHIKV have been regularly recorded for almost half a century. The first officially recognized outbreak in the country occurred in Samarinda, East Kalimantan in 1973 [12]. Since this time, reports of isolated CHIKV outbreaks in Indonesia have become more frequent. These reports reached a peak during a nationwide epidemic, which occurred between 2009–2010, resulting in 137,655 cases and dwarfing past case counts, which had never previously exceeded 10,000 a year [12]. Since 2011, official CHIKV case rates in Indonesia have returned to levels similar to those reported prior to the epidemic (S1 Fig). However, sporadic outbreaks have continued to occur across the archipelago [13–15]. Notably, two studies screening symptomatic travellers returning to Taiwan (2006–2009) and Japan (2006–2016) [16,17], found individuals returning from Indonesia were the most common source of imported cases of CHIKV. This highlights Indonesia's potential as a source of CHIKV transmission for the region; as the presence of competent vector species in Taiwan and Japan means that there is potential for an imported case of CHIKV to lead to local transmission, such as that documented in Italy in 2007 [18].

Despite suffering decades of outbreaks, official data on CHIKV incidence in Indonesia appears to be neglected, as CHIKV was not included in either of the 2018 or 2019 Ministry of Health annual reports (S1 Fig). Additionally, the vast majority of CHIKV epidemiology studies that have taken place in Indonesia have been largely restricted to the major, central islands of Java and Bali. This focus has made it difficult to ascertain whether CHIKV epidemiology in these densely populated regions is comparable to other parts of the country.

In this study, we aimed to characterise the contribution of CHIKV to febrile illness across three major regional cities in the east (Batam, Riau Islands), north (Banjamarsin, South Kalimantan) and west (Ambon, Maluku) of the country. Serum samples from febrile patients that had tested negative for dengue virus were tested for evidence of CHIKV infection by serology and RT-PCR. RT-PCR positive samples, including an additional set of archived samples, were subjected to genomic sequencing, and the resulting sequences were used to examine CHIKV evolutionary dynamics both within the country and in relation to the other countries of Southeast Asia.

## Methods

### Ethics statement

The study protocol was reviewed and approved by Eijkman Institute Research Ethics Committee (EIREC) with approval No. 113/2017. Written informed consent was obtained from patients recruited for the study. Written consent from parents or legal guardians was obtained on behalf of minors.

### Study sites and patient recruitment

Individuals between the age of 6 months and 75 years, presenting with fever greater than 38˚C for less than 5 days were recruited for this observational study from three city hospitals (S1 STROBE checklist): RS Santa Elisabeth situated in the west of Indonesia (Batam, Riau Islands), RS Ansari Saleh in central Indonesia (Banjamarsin, South Kalimantan) and RS Haulussy in east Indonesia (Ambon, Maluku) between September 2017 and September 2019. Sites were selected for their similar population size and healthcare infrastructure capacity. The sites are the capitals and largest cities (> 500,000 inhabitants) in their respective regions: Batam in Riau province, Ambon in Maluku province, and Banjarmasin in South Kalimantan province and represent three distinct regions of Indonesia, as reflected in their time zones (Western

Indonesia Time, Central Indonesia Time, and Eastern Indonesia Time). Upon hospital admission, single 3 to 5 ml blood samples were taken during the acute phase. Sera were separated by centrifugation and kept frozen at -20˚C until further processing. Patient sera were tested for dengue virus (DENV) by RT-qPCR and those testing negative were tested for evidence of CHIKV infection by serology and RT-PCR.

## RT-PCR and ELISA

Viral RNA was extracted from 200 μL sera using the MagNA Pure LC Total NA extraction kit (Roche, Switzerland) before being subjected to an alphavirus-specific RT-PCR as previously described [19]. Negative and positive controls for PCR were included in each batch and an RNA internal control from the Simplexa Dengue assay (Focus Diagnostics, Cypress, CA, USA) was used to monitor the RNA extraction process and to detect RT-PCR inhibition as part of a separate assay for dengue virus testing. Single RT-PCR replicates were run due to limitations in resources. Serum specimens with adequate volume remaining following RT-PCR were tested in duplicate for CHIKV IgM and IgG by in-house ELISA using acetone-extracted CHIKV antigen (strain Ross) as previously described [20,21]. ELISA units were calculated using the formula:

$$\left( \frac{mean\ OD\ (test\ sample) - mean\ OD\ (Negative\ control)}{mean\ OD\ (Weak\ positive\ control) - mean\ OD\ (Negative\ control)} \right) \times 100$$

A result of ≥40 units was considered positive.

Statistical comparisons of age and sex ratios between sites were performed in R v3.6.3 [22] using the Kruskal-Wallis Rank Sum test and Pearson's Chi-squared test respectively. Binomial confidence intervals for the observed IgG seropositivity rates were calculated using the Hmisc package in R [23]. A binomial regression using a complementary log-log link was fitted to the data and associations between IgG status and patient age, gender and location were tested by ANOVA using likelihood ratios, with the 'car' package in R [24].

## Nanopore library preparation and sequencing

A single RT-PCR positive sample from this study and 24 archived samples collected during other recent Indonesian outbreaks were subjected to complete coding region sequencing (Table 1). Viral RNA was reverse transcribed, amplified using a multiplex PCR tiling method, and sequenced on the Oxford Nanopore Technologies (ONT) MinION platform as previously described [25]. Multiplex sequencing libraries were prepared using the ONT Native Barcoding kit (EXP-NBD103) and the ONT 1D Ligation Sequencing kit (SQK-LSK108) according to the manufacturer's instructions. The resulting libraries were purified, loaded onto FLO-MIN106 flow-cells and sequenced using MinKNOW software v1.13.1.

## Consensus sequence reconstruction and analysis

Raw FAST5 output files were base-called using the default settings of Guppy v3.1.5 (ONT), discarding reads with a q-score below 7. De-multiplexing was performed using Qcat v1.0.7 (ONT) with default settings and the de-multiplexed FASTQ sequences were aligned to the CHIKV RefSeq sequence (NC_004162.2) using BWA mem v0.7.17 (option -x ont2d) [26]. Consensus genome sequences were generated from alignment files using samtools as previously described [27]. Finally, the consensus genomes were aligned using MAFFT v7.427 [28] and any major discrepancies such as indels or multiple consecutive SNPs were manually verified or corrected by referring back to the BAM read alignment file, viewed using Tablet

**Table 1. Overview of newly sequenced CHIKV strains included in phylogenetic analysis.**

| ID | Origin | Year | Accession number |
|---|---|---|---|
| TMH-073 | Tomohon, Sulawesi | 2014 | MT591085 |
| TMH-092 | Tomohon, Sulawesi | 2015 | MT591083 |
| TMH-104 | Tomohon, Sulawesi | 2015 | MT591084 |
| JMB-164 | Jambi, Sumatra | 2015 | MT591101 |
| JMB-167 | Jambi, Sumatra | 2015 | MT591103 |
| JMB-172 | Jambi, Sumatra | 2015 | MT591092 |
| JMB-187 | Jambi, Sumatra | 2015 | MT591098 |
| JMB-209 | Jambi, Sumatra | 2015 | MT591095 |
| JMB-288 | Jambi, Sumatra | 2015 | MT591102 |
| JMB-308 | Jambi, Sumatra | 2015 | MT591094 |
| JMB-331 | Jambi, Sumatra | 2015 | MT591093 |
| JMB-334 | Jambi, Sumatra | 2015 | MT591097 |
| JMB-337 | Jambi, Sumatra | 2015 | MT591100 |
| JMB-351 | Jambi, Sumatra | 2015 | MT591099 |
| JMB-474 | Jambi, Sumatra | 2015 | MT591096 |
| 201610125 | Buleleng, Bali | 2016 | MT591089 |
| 201610127 | Buleleng, Bali | 2016 | MT591086 |
| 201610133 | Buleleng, Bali | 2016 | MT591087 |
| 201610136 | Buleleng, Bali | 2016 | MT591088 |
| TBN-003 | Tabanan, Bali | 2017 | MT591104 |
| TBN-017 | Tabanan, Bali | 2017 | MT591090 |
| TBN-103 | Tabanan, Bali | 2017 | MT591091 |
| TBN-623 | Tabanan, Bali | 2018 | MT591105 |
| TBN-657 | Tabanan, Bali | 2018 | MT591106 |
| AMB-041 | Ambon, Maluku | 2018 | MT591107 |

v1.19.09.03 [29]. All consensus sequences were uploaded to Genbank under accession numbers MT591083—MT591107 (Table 1).

A maximum likelihood (ML) phylogeny was constructed from an alignment of all full-length CHIKV sequences available from GenBank as of 9th January 2020, using IQTree v1.6.11, implementing modelfinder and 1000 repetitions of the ultrafast bootstrap approximation method [30–32]. A time measured, maximum clade credibility (MCC) phylogeny was also inferred from an alignment of 135 complete and partial (> 1000 nt) coding sequences belonging to the Asian genotype using BEAST v1.10.4 [33]. To construct the phylogeny, CHIKV genomic nucleotide sequences of appropriate length were retrieved from GenBank using the search term: "Chikungunya virus"[porgn] AND (biomol_genomic[PROP] AND ("1000"[SLEN]: "13000"[SLEN]), aligned using MAFFT and a draft ML phylogeny was constructed using IQTree as described above. Sequences clustered within the Asian genotype clade were extracted (n = 357) and the associated metadata was parsed from GenBank files using gbmunge (https://github.com/sdwfrost/gbmunge). For sequences with missing metadata, a manual search of the literature was performed to identify the year of collection and country of origin. Sequences were excluded if this information was not readily available. Identical sequences were removed from the dataset using CD-hit-est, by clustering those with 100% identity [34] and the Americas lineage clade was reduced to 5 representative sequences to allow a clearer visualisation of the overall phylogeny. Finally, the temporal signal of the dataset was assessed by plotting the tip-to-root distance against time of sampling in R v.3.6.3, in

the manner employed by TempEst [35], followed by removal of any outliers. BEAST analysis was implemented with a general time reversible (GTR) model using a discretized gamma distribution with four categories (G4) plus invariant sites, a relaxed lognormal molecular clock, and a GMRF Bayesian skyride prior for effective population size. The resulting ML and MCC phylogenies were visualized using the ggplot2, ggtree, ape, colorspace and treeio packages in R v3.6.3 [36–38]. Multiple sequence alignment files, BEAST XML, log and tree files are available at: 10.5281/zenodo.3891450. Raw sequence data have been deposited in ENA under study PRJEB39967. Newly generated consensus sequences have been deposited in GenBank and their accession numbers are presented in Table 1.

## Results

### RT-PCR and serological testing

A total of 479 sera were tested for the presence of CHIKV RNA by RT-PCR and anti-CHIKV IgM by ELISA: 80 from Ambon, 208 from Banjarmasin and 191 from Batam (S1 Table). RT-PCR revealed the presence of alphavirus RNA in the serum of a 3-year-old, female patient from Ambon (AMB-041). Subsequent Sanger sequencing confirmed the presence of CHIKV. IgM antibodies were detected in a single patient from Banjarmasin, a 38-year-old male. The same patient's serum was also positive for anti-CHIKV IgG antibodies.

Sera with sufficient volume remaining (n = 460) were tested for the presence of anti-CHIKV IgG antibodies by ELISA: 69 from Ambon (median age = 10 y.o.; inter-quartile range (IQR) = 5–23; male to female ratio (M:F) = 1.56), 207 from Banjarmasin (median age = 16 y.o.; IQR = 10–24 y.o.; M:F = 1.11) and 184 from Batam (median age = 13 y.o.; IQR = 4–26; M:F = 1.19). No statistically significant differences in age (p > 0.05) or gender ratio (p > 0.05) were observed between sites. In total, IgG antibodies were detected in sera from 11/460 patients, giving an overall prevalence of 2.4% (95% C.I. = 1.3–4.2%) (Table 2). IgG seroprevalence ranged from 1.4% in Banjarmasin (95% C.I. = 0.5–4.2%) to 3.3% in Batam (95% C.I. = 1.5–7.0%). Overall IgG seroprevalence increased with age, with the exception of patients older than 50 years old (0/22). However, no statistically significant association was observed between IgG status and age, gender or site (all p > 0.05). Considering individual sites; in Banjarmasin, all IgG positive sera (n = 3) were obtained from patients aged 26 or over, whereas in Batam, all

**Table 2. Age specific IgG seroprevalence rates for chikungunya virus in three Indonesian cities.**

| Site | | CHIKV IgG ELISA Results Stratified by Age Group[*] | | | | | | | | |
|---|---|---|---|---|---|---|---|---|---|---|
| | | **0–5** | **6–10** | **11–15** | **16–20** | **21–30** | **31–40** | **41–50** | **>50** | **Overall** |
| **Ambon** | No. | 0/21 | 0/14 | 0/13 | 1/3 | 0/6 | 1/2 | 0/4 | 0/6 | 2/69 |
| | % (95% CI) | 0.0 (0.0–15.4) | 0.0 (0.0–21.5) | 0.0 (0.0–22.8) | 33.3 (1.7–80.0) | 0.0 (0.0–39.0) | 50.0 (0.3–97.4) | 0.0 (0.0–49.0) | 0.0 (0.0–39.0) | 2.9 (0.8–10.0) |
| **Batam[#]** | No. | 1/54 | 1/28 | 1/17 | 1/10 | 2/39 | 0/27 | 0/6 | 0/2 | 6/184 |
| | % (95% CI) | 1.9 (0.1–9.8) | 3.6 (0.2–17.8) | 5.9 (0.3–27.0) | 10.0 (0.5–40.4) | 5.1 (1.4–16.9) | 0.0 (0.0–12.5) | 0.0 (0.0–39.0) | 0.0 (0.0–65.8) | 3.3 (1.5–7.0) |
| **Banjarmasin** | No. | 0/17 | 0/36 | 0/43 | 0/39 | 1/36 | 1/16 | 1/6 | 0/14 | 3/207 |
| | % (95% CI) | 0.0 (0.0–18.4) | 0.0 (0.0–9.6) | 0.0 (0.0–8.2) | 0.0 (0.0–9.0) | 2.8 (0.1–14.2) | 6.3 (0.3–28.3) | 16.7 (0.9–56.4) | 0.0 (0.4–31.5) | 1.4 (0.5–4.2) |
| **Overall** | No. | 1/92 | 1/78 | 1/73 | 2/52 | 3/81 | 2/45 | 1/16 | 0/22 | 11/460 |
| | % (95% C.I.) | 1.1 (0.1–6.0) | 1.3 (0.1–7.0) | 1.4 (0.1–7.4) | 3.8 (1.1.-13.0) | 3.7 (1.3–10.3) | 4.4 (1.2–14.8) | 6.3 (0.3–28.3) | 0.0 (0.0–14.9) | 2.4 (1.3–4.2) |

[*] Presented as the number of positive samples/total samples within each group.

[#] Age data was missing for one sample from Batam (BTM-086), but was still included in the overall statistics.

IgG positive sera (n = 6) were obtained from patients younger than 30 years old. However, neither trend was statistically significant (p > 0.05).

### Phylogenetic analysis of Indonesian CHIKV strains

The genetic diversity of 25 Indonesian CHIKV strains collected between 2014 and 2018 (Table 1) was assessed by phylogenetic analysis. The resulting phylogeny demonstrated that all 25 newly sequenced isolates belonged to the Asian genotype (Fig 1) together with the majority of sequences previously reported from Indonesia (120 / 130).

A maximum clade credibility tree was also constructed in order to assess the evolutionary dynamics of Asian genotype CHIKV. The resulting phylogeny revealed that the Asian genotype clade is currently composed of two major sub-clades (boxes 1 and 2, and boxes 4 and 5), which were estimated to have diverged in 2000 (95% HPD: 1998–2002) (Fig 2).

The first of these clades (boxes 4 and 5) contained the majority of Asian genotype strains that have been recently isolated outside of Indonesia from countries including Singapore between 2013–2016, the Philippines between 2012–2016, Korea in 2014, and the Pacific Islands between 2014–2016. A number of strains from Indonesia were also present in this clade, all of

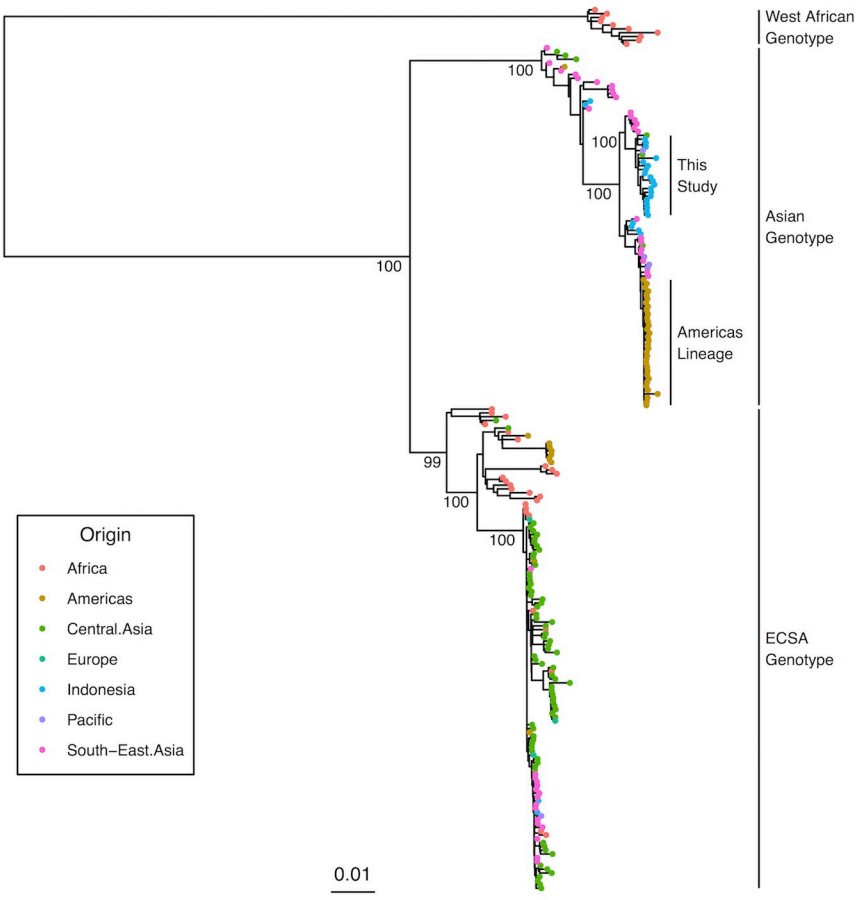

**Fig 1. Maximum likelihood phylogeny generated from full-length coding region sequences of Chikungunya virus.** Bootstrap branch support values are shown for major nodes of interest. Coloured tips denote each isolates' region of origin.

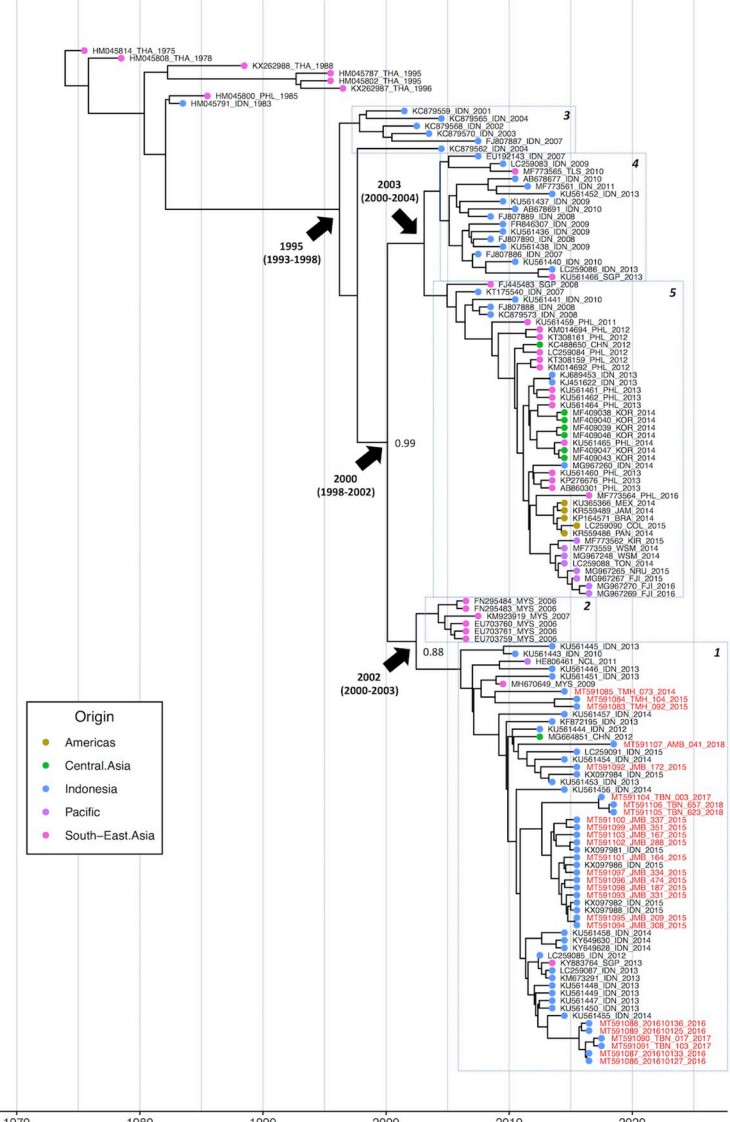

**Fig 2. Asian genotype chikungunya virus maximum clade credibility phylogeny.** The phylogeny was generated using all available Asian genotype strains, including a combination of full-length coding sequences and E1/E2 sequences (>1000 bp). Strains sequenced as part of this study are labelled in red. Colored tips denote the region of origin for each isolate. Major nodes of interest are labelled with their mean height and 95% highest probability density (HPD) intervals and posterior support values. The sub-clades, whose predicted amino acid sequences are compared in Table 3, are highlighted by dashed boxes and numbered accordingly. The Americas lineage included in box 5 was pruned to better visualise the overall phylogeny.

which were isolated between 2007–2014. The majority of these Indonesian isolates clustered into a distinct, but closely related group at the base of the clade. No strains belonging to this Indonesian group have been isolated since 2014.

More recent Indonesian strains all clustered within a second major clade (boxes 1 and 2), which predominantly contained isolates from Indonesia sampled in 2010 or later. Exceptions to this were a Malaysian isolate from 2009, a New Caledonian isolate from 2011, a Chinese isolate from 2012, and a Singaporean isolate from 2013. At the base of the clade, a cluster of Malaysian isolates formed a small, closely related group, estimated to have last shared a

common ancestor with the Indonesian strains in 2002 (95% HPD: 2000–2003). These Malaysian isolates were sampled during a nationwide outbreak in 2006, including one isolated from a macaque in 2007.

The samples sequenced in this study displayed a strong degree of similarity by region. Isolates from the 2015 outbreak in Jambi, Sumatra, were found to be closely related to each other, as well as to previously sequenced isolates from the same outbreak (KX097988, KX097981, KX097982). Similarly, isolates from the 2016 outbreak in Buleleng, Northern Bali, were also closely related to each other. The Buleleng outbreak strains appear to form part of an Indonesian lineage that has been repeatedly isolated between 2012–2014. These isolates include several cases from Bali in 2013 and 2014 (e.g. KM673291 and KY649630), and several strains isolated from Taiwanese travellers returning from undefined regions of Indonesia in the same years (e.g. KU561450 and KU561455). A single sequence deposited by researchers in Singapore was also present in this cluster, however the origin of this isolate is unclear. Two of the 2017 isolates from Tabanan in Southern Bali were also closely related to the Buleleng 2016 outbreak strain. However, a further three isolates sampled from Tabanan in 2017 and 2018 appeared to be more closely related to the Jambi 2015 outbreak strain, suggesting co-circulation of the two strains in Tabanan.

The three isolates collected from the remote site of Tomohon, North Sulawesi, in 2014–15, grouped together to form part of a small sub-cluster at the base of the main Indonesian clade. This sub-cluster, which also contained a strain isolated from Malaysia in 2009, was estimated to have diverged from the wider Indonesian lineage in 2006 (95% HPD: 2005–2007).

The single isolate from Ambon on Maluku Island, also clustered separately from most other samples sequenced in this study. The closest relatives to the Ambon strain were several strains detected in Taiwanese tourists returning from unspecified locations in Indonesia in 2014 and 2015, as well as two isolates sampled during the Jambi 2015 outbreak, one of which was sequenced as part of the current study (JMB-172). These two isolates from Jambi clustered separately from the vast majority of Jambi isolates, suggesting they were not the primary strain responsible for the 2015 outbreak in the city.

## Amino acid differences between Asian genotype CHIKV clades

In order to compare predicted amino acid differences between Asian genotype clades, 24 strains were selected to represent five distinct phylogenetic groups, which are highlighted by boxes in Fig 2. These groups corresponded to: 1) strains from the clade currently circulating in Indonesia; 2) Malaysian strains ancestrally related to the current Indonesian clade; 3) strains at the ancestral root of the entire Asian genotype clade; 4) older Indonesian strains closely related to strains currently circulating outside of Indonesia and; 5) strains from the Asian genotype clade in circulation outside of Indonesia, including strains from The Philippines, Korea, the Pacific Islands and the Americas.

Predicted translational differences between Asian genotype clades were observed in 6 of the 9 CHIKV genes (Table 3). Three of these predicted differences were identified as specific to the current Indonesian clade: a 7 amino acid deletion in the nsP3 gene ($nsP3_{376-382}$), a glutamine to arginine substitution at position 307 of the E2 gene (E2-Q307A), and an alanine to threonine substitution at position 321 of the E1 gene (E1-A321T). These differences were present in all strains of the current Indonesian clade except for the strains isolated in Tomohon. The Tomohon strains shared the 7 amino acid deletion $nsP3_{376-382}$ but retained an alanine at position 321 of the E1 gene, and displayed a unique E2-Q307H substitution in the E2 gene. The E1-321A residue observed in the predicted sequence of the Tomohon strains was also present in Asian genotype strains clustered outside of the current Indonesian clade, including

**Table 3. Overview of predicted amino acid differences between clades of Asian genotype CHIKV.**

Amino acid position (S27 strain)

| Group | Sample ID or accession number | Origin (* This study) | Collection date | Clade description | NSP1 121 | NSP1 226 | NSP3 376 | NSP3 377 | NSP3 378 | NSP3 379 | NSP3 380 | NSP3 381 | NSP3 382 | NSP3 383 | NSP3 437 | NSP3 457 | NSP3 483 | C 81 | E3 18 | E2 248 | E2 307 | E1 321 | E1 397 |
|---|---|---|---|---|---|---|---|---|---|---|---|---|---|---|---|---|---|---|---|---|---|---|---|
| 1 | 2016_10133 | Indonesia* | 2016 | Current Indonesian clade | A | T | - | - | - | - | - | - | - | I | A | T | D | M | Q | S | R | T | P |
| | KY883764 | Singapore | 2013 | | A | T | - | - | - | - | - | - | - | I | A | T | D | M | Q | S | R | T | P |
| | TBN_003 | Indonesia* | 2017 | | A | T | - | - | - | - | - | - | - | I | A | T | D | T | Q | S | R | T | P |
| | KX097986 | Indonesia | 2015 | | A | T | - | - | - | - | - | - | - | I | A | T | D | T | Q | S | R | T | P |
| | JMB_337 | Indonesia* | 2015 | | A | T | - | - | - | - | - | - | - | I | A | T | D | T | Q | S | R | T | P |
| | AMB_041 | Indonesia* | 2018 | | A | T | - | - | - | - | - | - | - | I | A | I | D | T | Q | S | R | T | P |
| | JMB_288 | Indonesia* | 2015 | | A | T | - | - | - | - | - | - | - | I | A | I | D | M | Q | S | R | T | P |
| | LC259091 | Indonesia | 2015 | | A | T | - | - | - | - | - | - | - | I | A | I | D | M | Q | S | R | T | P |
| | TMH_092 | Indonesia* | 2015 | | A | T | - | - | - | - | - | - | - | T | A | I | D | M | Q | S | H | A | P |
| 2 | FN295484 | Malaysia | 2006 | Malyasian relatives of the current Indonesian clade | A | T | - | - | - | - | - | - | - | I | A | I | D | M | Q | S | Q | A | P |
| | KM923919 | Malaysia | 2007 | | A | T | V | H | T | L | P | T | I | I | A | I | D | M | Q | S | Q | A | P |
| | EU703759 | Malaysia | 2006 | | A | T | V | H | T | L | P | T | I | I | A | I | D | M | Q | S | Q | A | P |
| 3 | HM045800 | Philippines | 1985 | Ancestors of the Asian genotype clade | A | T | I | H | T | L | P | T | T | I | A | T | D | M | Q | L | Q | A | P |
| | HM045814 | Thailand | 1975 | | A | T | V | H | T | L | P | T | T | I | A | T | D | M | Q | L | Q | A | L |
| | HM045808 | Thailand | 1978 | | A | T | V | H | T | L | P | T | T | I | A | T | D | M | Q | L | Q | A | L |
| | HM045791 | Indonesia | 1983 | | A | T | I | H | T | L | P | T | T | I | A | T | D | M | Q | L | Q | A | P |
| 4 | LC259083 | Indonesia | 2009 | Old Indonesian strains (relatives of current non-Indonesian clade) | A | I | I | H | T | - | - | - | - | T | A | I | D | T | Q | F | Q | A | P |
| | MF773561 | Indonesia | 2011 | | A | I | I | H | T | - | - | - | - | T | A | I | D | T | Q | F | Q | A | P |
| | LC259086 | Indonesia | 2013 | | A | I | I | H | T | - | - | - | - | T | A | I | D | T | Q | S | Q | A | P |
| 5 | LC259088 | Tonga | 2014 | Current non-Indonesian clade | E | I | I | H | T | - | - | - | - | T | T | I | N | T | R | F | Q | A | L |
| | MF773564 | Philippines | 2014 | | E | I | I | H | T | - | - | - | - | T | T | I | N | T | R | F | Q | A | L |
| | KR559489 | Jamaica | 2014 | | E | I | I | H | T | - | - | - | - | T | T | I | N | T | R | F | Q | A | L |
| | LC259090 | Colombia | 2015 | | E | I | I | H | T | - | - | - | - | T | T | I | N | T | R | F | Q | A | L |
| | KJ689453 | Micronesia | 2013 | | E | I | I | H | T | - | - | - | - | T | T | I | N | T | R | F | Q | A | L |
| Reference | NC_004162.2 | Tanzania | 1953 | Prototype strain S27 | A | T | I | H | T | L | P | S | A | T | V | T | N | T | Q | L | Q | A | L |

sequences ancestral to both clades. This is consistent with the results of the phylogenetic analysis, which placed the Tomohon strains at the base of the Indonesian clade (Fig 2).

The 7 amino acid deletion ($nsP3_{376-382}$) observed in every strain of the current Indonesian clade was confirmed in several isolates by Sanger dideoxy sequencing. Conversely, every strain from the non-Indonesian clade was found to possess a deletion of 4 amino acids at the same site ($nsP3_{379-382}$). This smaller deletion was also observed in several older Indonesian strains isolated between 2009–2013 but not in ancestral Asian genotype strains, which were isolated from Thailand, Indonesia and the Philippines in the 1970's and 1980's. The deletion in nsP3 was also absent from two of the three Malaysian strains isolated in 2006 and 2007. However, the presence of $nsP3_{376-382}$ in one of the Malaysian isolates suggests that the deletion occurred either prior to, or during, the Malaysian outbreak in 2006, in a strain that later gave rise to the new Indonesian lineage.

## Discussion

Molecular and serological testing of 479 febrile patients revealed that just 0.5% of cases (95% C.I. = 0.0–2.7%) across the three study sites were associated with CHIKV infection (RNA or IgM positive). In comparison, similar studies of febrile patients in a non-outbreak setting have reported CHIKV infection rates of 0.6% in Cambodia [39], 1.1% in Thailand [40] and 7.5% in the Philippines [41]. Within Indonesia, a prospective study, conducted at a single site in Bandung, central Java, between 2000–2004 and 2006–2008 reported CHIKV infections in 7.1% of febrile patients outside of any outbreak [42], while another recent multi-site study conducted in Java and Bali between 2013–16 reported a prevalence of 3.7% in febrile, dengue-negative patients [43] suggesting that, at present, CHIKV transmission is more common in these areas.

Overall, our study found that 2.4% (95% CI: 1.3–4.2%) of febrile patients across the three sites had previously been exposed to CHIKV. This is also relatively low in comparison to the results of a recent study of febrile patients in central Indonesia, which reported IgG seroprevalence rates ranging from 25.2–45.9% across 7 cities in Java, Bali and South Sulawesi [43]. High rates have also previously been reported for febrile patients from other countries in Asia, including the Philippines (57.5%), Vietnam (50.0%) and Sri Lanka (38.0%) [44]. However, a cross-sectional study of healthy adults (18–75 y.o.) in Singapore reported a similar rate to that observed here (2.2%), despite the country having experienced a CHIKV outbreak the previous year [45].

A recent meta-analysis of Indonesian studies reported an average IgG seropositivity rate of 14.1% (0.0% - 43.9%) for febrile patients in non-outbreak settings [12]. The majority of these studies were undertaken in Java and Bali. However, three reports from the sites of this study were included: in Ambon, a 1971 study described a similar IgG seroprevalence of 3.1% in febrile patients using a plaque-reduction neutralization test [46], while a study conducted the following year using a hemagglutination inhibition assay reported a seroprevalence of 11.5% [47]. A strong age-specific effect was observed in the second study; IgG seroprevalence increased from 2.7% in patients younger than 30, to 41.2% in those older than 30. Such a strong effect was not detected here, however a comparable seroprevalence of 1.7% was observed in patients younger than 30. It is possible that the overall difference in seroprevalence is due to no major CHIKV outbreaks having occurred in the region in the past 50 years. Indeed, patients involved in the 1972 study (0–9 y.o.), who now fall into the eldest two age groups in the present study (45–50+ y.o.), revealed no IgG positive cases in either instance. The same study also reported an IgG seroprevalence of 21.8% in febrile patients in Balikpapan, a city situated 500 km to the northeast of Banjarmasin. Seroprevalence was high across all age groups at this site, including 36% of individuals between 0–9 y.o. and was significantly higher

than other sites included in the study. Such an observation could possibly be explained by an outbreak of CHIKV having recently occurred in the region. In the present study, all 3 IgG positive cases in Banjarmasin were detected in adults (26–41 y.o). However, none were old enough to have participated in the previous study (>46 y.o.). Only 16 patients fell into this age range, indicating that additional sampling may be required for a more robust assessment of seroprevalence in older age groups.

The low seroprevalence observed in Ambon, Batam and Banjarmasin compared to cities in central Indonesia [43] may be due to an absence of recent outbreaks in the study regions. A similar regional difference has been reported from other Southeast Asian countries where the virus is endemic, including Thailand, where IgG seroprevalance was observed to be significantly lower in participants younger than 30 y.o. in the centre of the country (5.0%) compared to the south (15.6%), which had experienced two outbreaks in the past 20 years [48]. Study settings may also affect the reported prevalence, as a cross-sectional study of healthy adults (>35 y.o.) throughout Malaysia described a significant in difference in IgG seroprevalence between urban (7.1%) and rural (10.36%) dwelling participants [49].

Phylogenetic analysis of 25 newly generated sequences has demonstrated that the Asian genotype of CHIKV remains the dominant clade in Indonesia, despite a brief incursion by the ECSA genotype between 2008–2011. Further analysis of the Asian genotype clade revealed that the lineage currently circulating in Indonesia is genetically distinct from Filipino strains, as well as from those circulating in Indonesia prior to 2014. Strains from the current Indonesian lineage appear to have emerged in Indonesia between 2010–2013 and have been almost exclusively isolated from Indonesia and Malaysia, suggesting that the lineage is endemic to these two neighbouring countries. Unfortunately, no Asian genotype CHIKV sequences have been reported from Malaysia since 2009, and so it is not possible to ascertain whether strains from this lineage have remained in circulation there as they have in Indonesia.

An examination of the predicted amino acid sequences revealed several non-synonymous differences between the two main Southeast Asian clades of CHIKV. Specifically, a 7 amino acid deletion in the nsP3 gene, and E2-Q307R and E1-A321T residues in the structural genes defined the Indonesian clade. While the 7 amino acid deletion was present in all Indonesian isolates examined, the substitutions in the E1 and E2 genes were absent in a single strain isolated from Tomohon in 2015. Many of the predicted amino acid differences found in the Indonesian clade isolates were also observed in the closely related Malaysian strains from 2006–2007. These strains were isolated from both humans and macaques, and slight amino acid differences between isolates from these two hosts have been previously described [50]. All of the Indonesian strains sequenced as part of this study displayed the amino acid residues associated with a human host, namely: an arginine residue at position 221 of the nsP1 gene and the 7 amino acid deletion in the nsP3 gene.

It is unclear what effect, if any, the predicted differences in the nsP3, E1 and E2 genes might have on the virus. The 7 amino acid deletion of nsP3 was located in the C-terminal hyper-variable domain (HVD), which frequently contains significant deletions that appear to have no effect on the virus. However, this domain is involved in a high number of within-cell, host-protein interactions, which has led to suggestions that it may be involved in adaptation of the virus to new hosts [51]. Given the absence of the deletion in otherwise similar macaque isolates in Malaysia, one could speculate that it may be the result of zoonotic emergence from macaques into humans. The relevance of two further predicted differences, E1-T321A and E2-Q307R, in the envelope sequences of all strains within the current Indonesian clade, except a single strain from Tomohon has not been studied. However the E1 and E2 genes code for the two major surface glycoproteins of CHIKV, and are closely associated with transmissibility of the virus [52].

The Indonesian strains sequenced as part of this study largely clustered according to their region of origin. However, multiple strains were found to be in co-circulation in both Jambi in 2015, and Tabanan in 2017. There was also evidence of transmission occurring between geographically distinct regions: a close relative of one of the strains circulating in Jambi, Sumatra, in 2015 was later detected in Tabanan, Bali in 2017, while a distinct Jambi strain was related to the strain isolated from Ambon several years later. This finding possibly highlights Jambi as an important source of CHIKV transmission to other parts of Indonesia. However, this interpretation should be treated with caution, as, without knowing patient travel history, we cannot be certain as to whether infections were acquired locally or elsewhere.

Unsurprisingly, outbreak strains such as those in Jambi were highly inter-related, likely due to the rapid transmission that occurs during local epidemics. However, it is interesting to note that close relatives of the strain responsible for the 2016 outbreak in Buleleng, Bali, had been reported from the island 2 years earlier. Close relatives of the same strain were also detected in Tabanan, southern Bali in 2017. This finding suggests that additional, non-virological factors played a role in the Buleleng outbreak. Indeed, an entomological survey in Buleleng following the outbreak reported a high percentage of houses contained mosquito egg-laying sites (55%) [15], suggesting that increased vector density played a major role.

Several limitations to our study should be considered. Firstly, this work formed part of a wider investigation into the transmission dynamics of arbovirus infection in Indonesia. The sampling methods and sample sizes were therefore designed to capture cases of active infection, rather than robustly assess seroprevalence. The targeted recruitment of symptomatic patients may have therefore lead to an underestimation of IgG seroposivity rates in the overall population, due to the reduced likelihood of sampling individuals with existing immunity to local endemic pathogens such as CHIKV. The limited recruitment of older individuals may also have been a result of this recruitment strategy and the low numbers of IgG positive patients recruited overall were not sufficient for a robust statistical analysis. Additional testing of convalescent sera may have resulted in higher seroprevalence rates by allowing time for the IgM and IgG responses to develop. Additionally, the Asian genotype of CHIKV has been reported to cause relatively mild disease in comparison to other genotypes, particularly in young adults [42]. Therefore, it is possible that some CHIKV cases were excluded due to the recruitment criteria and the true prevalence of CHIKV at the study sites may be higher than observed.

Secondly, additional sequence data from a broader selection of countries throughout the region would allow a more confident interpretation of the current relationship between the strains of Asian genotype CHIKV in circulation. In particular, more data is needed from Southeast Asian countries such as Malaysia, Timor-Leste, the Philippines and Vietnam, where the virus is known to be endemic. This need for additional data also extends to Indonesia. Despite its current status as the most prolific contributor of CHIKV sequences in the region, the vast majority of Indonesian CHIKV sequences have been obtained from the main islands of Java and Bali. These two islands represent just 7 of the 34 provinces in the country, leaving large parts of the country under-represented. Given that Indonesia has been repeatedly reported as a major source of imported CHIKV cases across Asia [16,17], it follows that more detailed information from the country could play a major role in better understanding the transmission dynamics of CHIKV in the region.

In conclusion, CHIKV appears to have been a rare cause of febrile illness in the Indonesian cities of Ambon, Banjarmasin and Batam between 2017 and 2019. Further, the low prevalence of anti-CHIKV IgG observed across all age groups at each of the three sites, suggests few individuals in these regions have been exposed to CHIKV in recent decades. Phylogenetic analysis of several recently isolated Indonesian strains revealed that all belong to a closely related sub-

clade within the Asian genotype. Strains of this sub-clade appear to have recently replaced the previous lineage in Indonesia, which can still be found circulating in several countries of Southeast Asia, the Pacific Islands and the Americas. Given consistent reports of imported cases of CHIKV from Indonesia to neighbouring countries, it is possible that this clade coukd be introduced to other countries in the region.

## Supporting information

**S1 STROBE Checklist.**
(DOC)

**S1 Fig. Official reports of chikungunya virus cases by the Indonesian Ministry of Health.**
No data on chikungunya virus were included in the 2018 or 2019 reports. Data extracted from annual reports, accessed at: pusdatin.kemkes.go.id/folder/view/01/ structure-publikasi-pusda-tin-profil-kesehatan.html, on 1st June 2020.
(TIFF)

**S1 Table. Age and gender demographics of study participants.**
(XLSX)

## Acknowledgments

The authors wish to thank Drs Yanuarni Pamai, Anna Afida, Ingrid Hutagalung, Kartika Sari, Ni Putu Diah Witari, Dekrit Gampamole, and Sotianingsih Haryanto for their assistance in patient recruitment and sample collection; Yora Permata Dewi for performing the ELISA testing; and Professor Andres Merits (University of Tartu, Estonia) for kindly providing us with control material to test the CHIKV sequencing methodology.

## Author Contributions

**Conceptualization:** Samuel C. B. Stubbs, R. Tedjo Sasmono, Khin Saw Aye Myint, Simon D. W. Frost.

**Data curation:** Samuel C. B. Stubbs, Edison Johar, Frilasita A. Yudhaputri, Benediktus Yohan, Marsha S. Santoso, Rahma F. Hayati, Dionisius Denis.

**Formal analysis:** Samuel C. B. Stubbs.

**Funding acquisition:** Barbara A. Blacklaws, R. Tedjo Sasmono, Khin Saw Aye Myint, Simon D. W. Frost.

**Investigation:** Edison Johar, Frilasita A. Yudhaputri, Benediktus Yohan, Marsha S. Santoso, Rahma F. Hayati, Dionisius Denis.

**Project administration:** Edison Johar, Frilasita A. Yudhaputri, Benediktus Yohan, Marsha S. Santoso, Rahma F. Hayati, Dionisius Denis.

**Resources:** Ann M. Powers.

**Supervision:** Barbara A. Blacklaws, R. Tedjo Sasmono, Khin Saw Aye Myint, Simon D. W. Frost.

**Visualization:** Samuel C. B. Stubbs.

**Writing – original draft:** Samuel C. B. Stubbs.

**Writing – review & editing:** Barbara A. Blacklaws, Ann M. Powers, R. Tedjo Sasmono, Khin Saw Aye Myint, Simon D. W. Frost.

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
