## [Decision Letter · Decision Letter 0]

18 Aug 2020

Dear Mr Stubbs,

Thank you very much for submitting your manuscript "An investigation into the epidemiology of chikungunya virus across neglected regions of Indonesia" for consideration at PLOS Neglected Tropical Diseases. As with all papers reviewed by the journal, your manuscript was reviewed by members of the editorial board and by several independent reviewers. In light of the reviews (below this email), we would like to invite the resubmission of a significantly-revised version that takes into account the reviewers' comments. 

We cannot make any decision about publication until we have seen the revised manuscript and your response to the reviewers' comments. Your revised manuscript is also likely to be sent to reviewers for further evaluation.

Sincerely,

Marilia Sá Carvalho

Associate Editor

Elizabeth Carlton

Deputy Editor

Reviewer's Responses to Questions

**Key Review Criteria Required for Acceptance?**

**Methods**

-Are the objectives of the study clearly articulated with a clear testable hypothesis stated?

-Is the study design appropriate to address the stated objectives?

-Is the population clearly described and appropriate for the hypothesis being tested?

-Is the sample size sufficient to ensure adequate power to address the hypothesis being tested?

-Were correct statistical analysis used to support conclusions?

-Are there concerns about ethical or regulatory requirements being met?

Reviewer #1: The study aims to characterize the contribution of CHIKV to febrile illness across three cities from Indonesia. The authors had ethical approval to collect samples from symptomatic patients. Despite obtaining 482 samples, the authors were only able to detect CHIKV RNA in only one sample from this recent study-population, besides the detection of IgG in 11 cases. To assess the evolutionary dynamics of CHIKV in Indonesia, the authors performed next-generation sequencing on the single sample identified plus the other 25 archived samples also from previous outbreaks in Indonesia.

Reviewer #2: Yes

Reviewer #3: Line 132-134: Please describe the IgG ELISA and any validation done for the assay. I could find limited description about this in the references (much about IgM assay) and please state here what viral strain/protein was used in the ELISA.

**Results**

-Does the analysis presented match the analysis plan?

-Are the results clearly and completely presented?

-Are the figures (Tables, Images) of sufficient quality for clarity?

Reviewer #1: Results were in general well described and matched the analysis plans. however, it seems there is an inconsistency in lines 231-232 when the authors say that strains from the Indonesia clade were collected between 2007 and 2013 and claim that no further strains from that clade were isolated since 2013. A closer look at the MCC tree box 5 shows an Indonesian isolate sampled in 2014 (MG967260_IDN). Therefore authors should clarify or rectify their statement in the text.

Reviewer #2: Yes

Reviewer #3: Line 180: It would be helpful to see a figure of the full age distribution of the participants in each place. This is important for interpreting the seroprevalence results. Please comment on the age of the IgG seropositive cases, if there is any pattern with age, then it may be important to age standardise the reported seroprevalences. 

The paper does however seem to be somewhat skewed towards the results in this section in the main text, with less attention paid to important more detailed analysis/presentation of analysis of the PCR and seroprevalence results.

The data availability statement says the data is available, but I could not find the link to this- please add in. If possible this should be list of samples tested with age and then the results of each of the assays. If not possible, grouped into as small age groups as possible.

**Conclusions**

-Are the conclusions supported by the data presented?

-Are the limitations of analysis clearly described?

-Do the authors discuss how these data can be helpful to advance our understanding of the topic under study?

-Is public health relevance addressed?

Reviewer #1: Conclusions and study's limitations were described, although more data are necessary to support the final statement in the conclusion section (lines 429-430).

Reviewer #2: Yes.

Reviewer #3: Line 331: This comment on age results in the current study needs to be backed up by more presentation of the age seroprevalence results in the text. 

Line 332: Please expand on the age comparison between current and past studies. Another relevant comparison is between the age groups that would have been included in both studies (but 40 years older in the latest study) and how this compares to your results. 

Line 333: I’m not at all clear what the < 30 years olds seroprevalence is from past and this study currently. Please clarify. 

Line 340: What is the data you are comparing to from Java and Bali? Even if quoted elsewhere please restate here. 

Limitations- how do you think the choice of symptoms in subjects could have influenced the PCR and serology results.

**Editorial and Data Presentation Modifications?**

Reviewer #1: (No Response)

Reviewer #2: The manuscript is well written - it could possibly be presented somewhat better by not having so many very short paragraphs, but that is a minor concern.

Reviewer #3: (No Response)

**Summary and General Comments**

Reviewer #1: (No Response)

Reviewer #2: The manuscript is very straightforward, reporting on new Asian genotype sequences of CHIKV as well as a serological study on febrile patients in regional areas of Indonesia. Studies that add to our understanding of patterns of CHIKV circulation are very valuable, given how neglected this virus is and the sporadic nature of outbreaks. 

There is a limitation here in that serological surveys should be much larger in sample size rather than simply opportunistic like this but it is a question of resources. They should also span a broader range of the population in age, not just febrile patients who overall were very young. Given that CHIKV has such large intervals between outbreaks, it's possible that a skew towards very young patients is likely to miss serological patterns in older age groups. I felt that the authors dealt with this limitation relatively well in their discussion, but this could be emphasised. 

Minor comments:

Abstract – lines 30-31: please specify that the archived samples were from outbreaks across the whole of Indonesia so it’s clearer to the reader

Line 66: it is misleading to claim the Asian genotype ‘emerged’ in the Americas; a better phrase would be ‘was introduced’.

Lines 132 to 133: were negative PCRs run in duplicate? Were there any controls for RNA extraction? How do we know these are true negatives? 

Line 135: binomial confidence intervals for what? 

Lines 166: how were these sequences for Figure 2 chosen? It does not seem like all Asian genotype CHIKV sequences reported to have an Asian origin were included. At least it’s not clear if all were included or not. 

Line 187: please clarify what was being tested between groups for significance

Line 416 – 417: can the authors please provide a reference? 

The discussion could be a little bit broader in its perspective on the Asian genotype's history in Asia, rather than only focusing on Indonesia. 

The authors state the sequencing data will be banked at European Nucleotide Archive but I would have expected some provisional accession numbers to be reported. I hope these are provided with any revised version.

Reviewer #3: This an interesting paper, that adds to the picture of chikungunya transmission in Indonesia. The presentation of the seroprevalence results is too brief, and further analysis with regards to age. The discussion of these results in the context of serosurveys from elsewhere is also not sufficiently clear or detailed. 

Abstract: 

Conclusion: Please state which of your results leads you to conclude low-level endemic transmission. 

Author summary: 

I think the statement that this lineage is likely to be introduced is a little strong. Perhaps will lead to imported cases, but whether that will lead to virus introduction, depends on the factors of the countries it is imported to. 

Main text: 

Line 71-73: Please more details on the countries across the region where chik is endemic, and the definition of endemic. 

Line 90: Suggest more details on where these studies were, what their transmission potential for chik is, and how it was determined that Indonesia was the source of these imported cases. 

Line 91: Same comment as the author summary, consider whether there is sufficient evidence that Indonesia is a source for transmission in other countries. 

Line 102-103: Please expand on how it was determined to chose these sites and that they are representing the regions, rather than are in each region.

PLOS authors have the option to publish the peer review history of their article (what does this mean?). If published, this will include your full peer review and any attached files.

Reviewer #1: No

Reviewer #2: No

Reviewer #3: No
---

## [Decision Letter · Decision Letter 1]

30 Oct 2020

Dear Mr Stubbs,

We are pleased to inform you that your manuscript 'An investigation into the epidemiology of chikungunya virus across neglected regions of Indonesia' has been provisionally accepted for publication in PLOS Neglected Tropical Diseases.

Best regards,

Marilia Sá Carvalho

Associate Editor

Elizabeth Carlton

Deputy Editor

Reviewer's Responses to Questions

**Key Review Criteria Required for Acceptance?**

**Methods**

-Are the objectives of the study clearly articulated with a clear testable hypothesis stated?

-Is the study design appropriate to address the stated objectives?

-Is the population clearly described and appropriate for the hypothesis being tested?

-Is the sample size sufficient to ensure adequate power to address the hypothesis being tested?

-Were correct statistical analysis used to support conclusions?

-Are there concerns about ethical or regulatory requirements being met?

Reviewer #3: (No Response)

**Results**

-Does the analysis presented match the analysis plan?

-Are the results clearly and completely presented?

-Are the figures (Tables, Images) of sufficient quality for clarity?

Reviewer #3: (No Response)

**Conclusions**

-Are the conclusions supported by the data presented?

-Are the limitations of analysis clearly described?

-Do the authors discuss how these data can be helpful to advance our understanding of the topic under study?

-Is public health relevance addressed?

Reviewer #3: (No Response)

**Editorial and Data Presentation Modifications?**

Reviewer #3: (No Response)

**Summary and General Comments**

Reviewer #3: (No Response)

PLOS authors have the option to publish the peer review history of their article (what does this mean?). If published, this will include your full peer review and any attached files.

Reviewer #3: No

---

## [Editor Report · Acceptance letter]

28 Nov 2020

Dear Mr Stubbs,

We are delighted to inform you that your manuscript, "An investigation into the epidemiology of chikungunya virus across neglected regions of Indonesia," has been formally accepted for publication in PLOS Neglected Tropical Diseases.

Best regards,

Shaden Kamhawi

co-Editor-in-Chief

Paul Brindley

co-Editor-in-Chief
